# HPAEC-PAD Analytical Evaluation of Carbohydrates Pattern for the Study of Technological Parameters Effects in Low-FODMAP Food Production

**DOI:** 10.3390/molecules28083564

**Published:** 2023-04-19

**Authors:** Olimpia Pitirollo, Maria Grimaldi, Claudio Corradini, Serena Pironi, Antonella Cavazza

**Affiliations:** 1Dipartimento di Scienze Chimiche della Vita e della Sostenibilità Ambientale, Università di Parma, Parco Area delle Scienze 17/A, 43124 Parma, Italy; 2Dipartimento di Ingegneria e Architettura, Università di Parma, Parco Area delle Scienze 181/A, 43124 Parma, Italy; 3BRU.PI srl, Via Berlino, 91, 47822 Santarcangelo di Romagna, Italy

**Keywords:** FODMAPs, HPAEC-PAD, fructo-oligosaccharides, flours, fermentation, baking

## Abstract

Background: “FODMAPs” (fermentable-oligo-, di-, monosaccharides, and polyols) are a group of fermentable carbohydrates and polyols largely diffused in food products. Despite their beneficial effects as prebiotics, people affected by irritable bowel syndrome manifest symptoms when eating these carbohydrates. A low-FODMAP diet seems to be the only possible therapy proposed for symptom management. Bakery products are a common source of FODMAPs, whose pattern and total amount can be affected by their processing. This work aims at studying some of the technological parameters that can influence the FODMAPs pattern in bakery products during the production process. Methods: high-performance anion exchange chromatography coupled to a pulsed amperometric detector (HPAEC-PAD) was used as a highly selective system for carbohydrates evaluation analyses on flours, doughs, and crackers. These analyses were performed using two different columns, the CarboPac PA200 and CarboPac PA1, which are selective for oligosaccharide and simple sugar separation, respectively. Results: emmer and hemp flours were selected to prepare doughs as they contained low oligosaccharide content. Two different mixes of ferments were used at different times of fermentation to evaluate the best conditions to achieve low-FODMAP crackers. Conclusion: the proposed approach allows carbohydrate evaluation during crackers processing and permits the selection of opportune conditions to obtain low-FODMAP products.

## 1. Introduction

Fibers are composed of oligosaccharides and polysaccharides occurring in food of plant origin and are resistant to human enzymatic digestion. They can be distinguished into soluble fibers, such as fructo-oligosaccharides (FOS), inulins, pectins, and insoluble fibers, which include high molecular weight polysaccharides, such as cellulose [1]. Soluble fibers absorb water in the intestinal tract, making a viscous and gelatinous substance that causes a slow transit of the bolus in the intestine, increasing the sense of satiety and decreasing the absorption of some substances such as cholesterol and glucose [2]. Furthermore, the soluble fibers show prebiotic activity, promoting the growth of lactic bacteria [2].

Fructans belong to the class of soluble fibers, they are complex molecules formed by a sucrose residue elongated with fructose monomers linked by β (2→1) bond. Fructans with a low degree of polymerization (DP3-DP10) are commonly named fructo-oligosaccharides (FOS), which include 1-kestose (DP3), 1-nystose (DP4), and fructosyl-nystose (DP5). Among fructans with long chains, there is inulin, a linear polydisperse polymer with different polymerization degrees (DP10-DP60) [3].

Another class of compounds belonging to soluble fibers is galacto-oligosaccharides (GOS). They are constituted by one molecule of glucose (Glu) and one to eight units of galactose (Gal), which are divided into α-galacto-oligosaccharides (α-GOS) and β-galacto-oligosaccharides (β-GOS) [4].

FOS, GOS, fructose, and polyols belong to a class of compounds defined as FODMAP (fermentable oligo-, di-, monosaccharides, and polyols) [5]. FODMAPs are characterized by prebiotic activities exerted as they are not absorbed in the colon, and therefore become a good substrate for the growth of bacteria in the microbiota [6]. However, recent studies show that the fermentation of these molecules negatively affects subjects suffering from inflammatory diseases, such as gastrointestinal disorders [7]. In particular, irritable bowel syndrome (IBS) is a common gastrointestinal disorder affecting 5–12% of the population [8]. The physiopathology of IBS is not clear because it can show different symptoms among patients, but includes abdominal pain, bloating, and flatulence [8]. The implication of different physiopathology processes makes IBS a disorder with heterogeneous features, and the lack of knowledge about the enteric nervous system does not help the development of drug therapies [9]. As a possible alternative, medical nutritional advice includes the administration of probiotics and a gluten-free diet (even if patients have not been diagnosed with celiac disease or suffering from gluten sensitivity) [10] low in fiber, fat, and nerve substances, such as any gastrointestinal inflammation [11].

Diet is therefore the main IBS treatment, and at present, a low-FODMAP content diet is suggested to control symptomatology [11]. It is reported that the original description of a low-FODMAP diet was done in Australia ten years ago [9], and today it has been internationally accepted for the treatment of IBS because it has led to symptom improvement in about 76% of patients [12]. The low-FODMAP diet requires knowledge of the FODMAP levels in foods in order to exclude those giving negative effects. However, the low-FODMAP diet is a protocol to be followed for a limited period of time in order to allow the individual to understand the tolerated quantities of one or more FODMAPs, with the aim of scheduling a personalized food plan. A long term diet without a balanced food plan can lead to a lack of fibers (and therefore a potential increase in dysbiosis), mineral salts, vitamins, and an imbalance of the main nutritional principles [13].

FODMAPs are present in a wide range of foods, including fruits and vegetables, wheats, cereals, legumes, nuts, seeds, dairy products, meat, fish, fats, oils, drinks, and their derivatives [14]. In addition to the knowledge of the composition of a food, it is important to know the cut-off values of FODMAPs in the products in order to draw up a correct diet. The cut-off values were determined considering the FODMAPs content and the typical portion size of foods consumed in a meal that commonly trigger symptoms in individuals with IBS. It corresponds to 0.5 g of total FODMAPs per sitting, and not more than 3 g per day [11]. This made it possible to establish threshold levels for each FODMAP above which most people have symptoms. Monash University developed a 297-point questionnaire called the “Comprehensive Nutrition Assessment Questionnaire (CNAQ)” to analyze the consumption of micro/macronutrients and FODMAPs in a patient’s diet [11].

Wheats and their derivatives are the main sources of FODMAPs in diet [15]. It is reported that cereal products may contain several types of short-chain carbohydrates, and their amount depends on the nature of the cereal used in the manufacture of these products [14]. For example, rice products contain lower FODMAP content compared with that of wheat or rye products [5]. The FODMAP levels in products depend on the choice of the type of ingredients but are also affected by technological parameters selected during the manufacturing process, including the type of yeast/ferment used, the length of fermentation [15,16], and also the cooking conditions [17]. A possible approach to obtain low-FODMAP products is to investigate the effects of those parameters and to adjust them in order to limit oligosaccharides’ occurrence. In fact, the degradation of fructans mainly occurs during fermentation, providing a reduction of oligosaccharides; in addition, the cooking step can have an impact, because fructans are susceptible to thermal degradation at high temperatures. Consequently, the fructan content is lower after cooking; however, at the same time, the amount of sucrose, fructose, and glucose increases because they are released as degradation products [18]. The aim of this work is to present a methodology able to investigate the effects of some of the technological parameters influencing the FODMAP content in final bakery products by using high-performance anion exchange chromatography with a pulsed amperometric detector (HPAEC-PAD), which is a highly selective analytical tool for carbohydrates characterization [19]. The knowledge of the phenomena involved during the different technological steps, affecting sugars amount, is mandatory for achieving low-FODMAP products. In fact, not only is the selection of raw materials an important phase, but also the choice of ferments, the time of fermentation, and the baking process can affect the final FODMAP content.

## 2. Results and Discussion

### 2.1. Carbohydrates Analysis by HPAEC-PAD

The selected extraction method of carbohydrates from samples was based on a previous article reported by Ziegler et al. [8] that considered the use of methanol, necessary to inhibit the α-amylase activity, and preventing starch hydrolysis during the extraction process. Ispiryan et al. [20] determined the α-amylase activity by HPAEC-PAD in whole wheat flour extracted in different conditions. Indeed, the authors found that the chromatograms of extracts obtained without methanol treatment contained large amounts of glucose and fructose and less amount of sucrose, whereas the extracts treated with methanol presented a large amount of sucrose and less amount of glucose and fructose. The use of methanol is thus necessary to avoid overestimation of glucose, which can lead to a misinterpretation of the fructose/glucose ratio, which is important for determining whether fructose is considered a FODMAP.

The analysis of standards was achieved under the selected chromatographic conditions using the CarboPac PA200 column for oligosaccharides and the CarboPac PA1 column for simple sugars separation, employing the gradients reported in Table A1 and Table A2 of Appendix A (Figure A1 and Figure A2, Appendix A). LODs and LOQs were determined for all available standards (Table A3, Appendix A). The precision of retention time of all standards was evaluated intra-day (n = 10) and inter-day (n = 30), resulting in RSD% < 3% (Table A3, Appendix A). The precision of the peak area of all standards was calculated intra-day (n = 10) and inter-day (n = 30) for the reference concentration of 1 μg/mL, and the resulting RSD% was <3%.

Calibration curves were built with commercially available standards to perform quantitative analyses, and all of them presented a good linearity, with R2 > 0.99 (Table A4, Appendix A) in the range of analyzed concentrations (0.25–10.0 μg/mL).

To evaluate if the extraction method was exhaustive, a comparison of the results obtained by the analysis of pre-spike and post-spike samples was assessed using the internal standard L-rhamnose monohydrate at a concentration of 1 μg/mL. Pre-spike samples (three replicates) were prepared by adding the internal standard before the extraction procedure, whereas in the post-spike samples, the internal standard was added after the extraction procedure. The comparison between the areas of the peaks obtained in the two samples did not show a statistically significant difference. The calculated recovery value was about 90%.

### 2.2. Selection of Flours

The choice of flour, which is the main ingredient of bakery products, is an important parameter to produce low-FODMAP foods because many carbohydrates already occur in the raw material. In this work, different flours from wheat, rice, emmer, and hemp were analyzed by employing a PA200 chromatographic column to check which of them were the lowest in oligosaccharides (FOS and/or GOS). The obtained chromatograms of flours reported in Figure 1 are divided into two sections. The first section ranges from 0 to 15 min of analysis where the elution of polyols, mono-, di-saccharides, 1-kestose, and 1-nystose occurs, which are the DP3 and DP4 of FOS, respectively (chromatogram of standards Figure A1, Appendix A). With this column, the polyols, mono-, and di-saccharides co-eluted and were not well separated (Figure A1, Appendix A). The second section, from 15 min onwards, shows the elution of oligosaccharides (DP > 4) such as FOS or GOS.

The limit of this technique is that the only commercially available standards of FOS are DP3 (1-kestose) and DP4 (1-nystose), therefore a punctual identification of other peaks eluting after DP4 cannot be confirmed (Figure 1), although their identity can be assigned considering the accepted assumption that each peak eluting at an equal distance from the previous one corresponds to a subsequent unit of DP [19].

Therefore, the recorded chromatograms can be of great help for general screening, and a semiquantitative approach can be proposed by evaluating peak areas of each oligosaccharide, allowing researchers to monitor the eventual variations of their amounts.

Consequently, the comparison between the chromatograms permits us to put in evidence the flours containing higher amounts of oligosaccharides based on the number of and the areas of the peaks eluting in the second part of the chromatograms in Figure 1.

Based on these considerations, from the chromatograms reported in Figure 1, it is possible to see that emmer and hemp flours have the lowest oligosaccharides content. As a consequence, emmer and hemp flours were analyzed by using the PA1 column—which is selective for simple sugar separation—in order to check the amounts of polyols and fructose compared with glucose. Indeed, fructose is considered a FODMAP sugar if its amount is higher than glucose [8]. Therefore, the fructose/glucose ratio is important to evaluate the amount of fructose compared with glucose. In the PA1 column, the polyols, mono-, and di-saccharides were well separated (Figure A2, Appendix A), and quantitative analysis was performed. The data collected for emmer and hemp flours are reported in Figure 2.

The data showed that hemp flour contains less simple sugars compared with emmer flour (Figure 2A). In detail, hemp flour contained 46% less glucose and 30% less fructose compared with emmer flour, whereas sucrose was not detected in hemp flour. The fructose/glucose ratio was >1 in both flours (Figure 2B), meaning that fructose was in excess compared with glucose. The fructose/glucose ratio was definitely higher in hemp flour. These two flours were selected and mixed together to produce different bakery products. However, it should be considered that the transformation process (dough preparation, fermentation step, and cooking process) can lead to differences in final products. In this phase, it was important to select flours having fewer peaks corresponding to oligosaccharides (FOS and or GOS) in order to minimize the fructose and polyols formation during the production process [21].

### 2.3. Mix of Yeast/Ferment to Reduce FODMAPs in Doughs

It is well known that the type of ferment and time of fermentation are important parameters to be considered in order to obtain low-FODMAP products [22]. In fact, during fermentation, the microbial colonies occurring in the dough are able to digest sugars, providing a number of degraded sugars that is directly proportional to the time of fermentation. However, to obtain low-FODMAP products, it is important to find an equilibrium between the type of ferments used and the time of fermentation. In fact, it is necessary to degrade FOS and GOS, but on the other side, it has to be taken into account that fructose and polyols increase [22] as a consequence during high-FODMAP bakery production. The analysis of simple sugars can be of help to evaluate fructans and GOS hydrolysis occurring during fermentation.

By choosing the right mix of ferments and the right time of fermentation, it is possible to find a compromise to ensure that each class of carbohydrates, in the final product, adheres to the total cut-off limit (0.5 g per portion) established by Monash University [11,22].

Two types of doughs were prepared by using a mixture of the flours selected in the previous step (emmer and hemp), and two different yeasts/ferments (named 1 and 2), which were left for different times of fermentation: 4, 6, and 8 h (Table 1).

The goal was to understand which was the best mix of yeast/ferment, and the optimal time of fermentation to obtain a dough that could ensure low-FODMAP final products.

Analysis by HPAEC-PAD using the CarboPac PA200 column was performed for qualitative analysis of oligosaccharides. A semi-quantitative evaluation was possible by measuring peak area values. In dough 2, peaks related to oligosaccharides were not detected at all times of fermentation. This means that ferment 2 used for dough 2 preparation was able to degrade oligosaccharides already present in flours after 4 h of fermentation. In dough 1, by increasing the time of fermentation, the number and intensity of peaks corresponding to oligosaccharides (FOS/GOS) slightly decreased.

The qualitative and quantitative analysis of simple sugars was performed using the PA1 column, and data are reported in Figure 3.

As shown in Figure 3, in dough 1 (Figure 3A), polyols, glucose, and fructose increased with the increase of time of fermentation (from 4 h to 8 h), whereas the amount of galactose was not significantly different from 4 to 8 h of fermentation. In addition, the fructose/glucose ratio was ≤1, thus allowing fructose to be classified as low-FODMAP.

Dough 2 (Figure 3B), after 6 h of fermentation, presented fewer polyols (sorbitol) than dough 2 after 4 and 8 h of fermentation; the amount of galactose was below the LOQ, except for dough 2 at 4 h of fermentation. The fructose/glucose ratio in dough 2 was >1, except after 4 h of fermentation.

It is possible to assume that both oligosaccharide types, FOS and GOS, were probably degraded during the fermentation time from the mix of yeasts/ferments. Indeed, glucose, fructose, and galactose have been identified in dough 1, and their amount increased with the time of fermentation (Figure 4A). In dough 2, a different degradation occurred for FOS than GOS because galactose was detected in an amount below LOQ, except for the dough at 4 h of fermentation (Figure 4B).

Finally, considering all the reported results carried out using the two different columns (CarboPac PA200 and CarboPac PA1) dough 1_6h and dough 2_6h were selected to prepare crackers. These doughs represent the best compromise between oligosaccharides, fructose/glucose ratio, and the amount of polyols (sorbitol) to ensure low-FODMAP final products.

### 2.4. Baking Processing and FODMAPs Content in Final Products

Because all steps of food processing are believed to influence the FODMAPs content in final products [23], the effects of baking on the carbohydrates pattern in crackers were also evaluated.

The selected dough 1 and dough 2 were used to prepare cracker 1 and cracker 2 after 6 h of fermentation. Chromatograms obtained with a CarboPac PA200 column of cooked crackers did not show oligosaccharide peaks.

Figure 4A shows the quantitative analysis of simple sugars (CarboPac PA1 column) of cooked crackers in comparison with the respective dough. In general, the amount of simple sugars increased after cooking processing for both crackers, cracker 1C, and cracker 2C prepared from dough 1–6 h and dough 2–6 h, respectively. In particular, fructose increased between 60–70% and glucose increased between 70–80% (Figure 4B), meaning that FOS has been degraded during the cooking processing, giving a fructose/glucose ratio <1. Galactose only occurred in cracker 1C, and its amount increased by about 35%, while sorbitol only by 6% compared with the corresponding dough (Figure 4B). In cracker 2C, sorbitol increased by 63% compared with the corresponding dough.

It is possible to affirm that the cooking process influences the FODMAPs amount in final products, which is also correlated to the mix of yeast/ferment used. Indeed, during the first phase of the cooking process, there is an overactivity of the ferments until the dough reaches the optimal temperature for yeast/ferment inactivation. In that time, oligosaccharides might be degraded, releasing simple sugars and polyols.

## 3. Materials and Methods

### 3.1. Chemicals

Water (MilliQ), sodium acetate, sodium hydroxide 50% *w*/*w*, xylitol, sorbitol, mannitol, rhamnose, glucose, galactose, fructose, sucrose, lactose, raffinose, maltose, 1-kestose, and 1-nystose analytical standards were purchased from Sigma Aldrich (Steinheim, Germany) with the highest purity degree (>99%). Inulin from chicory roots was purchased by Beneo (Mannheim, Germany).

### 3.2. Samples

The samples of analysis were four flours purchased on the market and doughs and crackers produced by BruPi srl (Sant’Arcangelo di Romagna, RN, Italy). A complete list of the analyzed samples is reported in Table 1.

Hemp and emmer flours were mixed together and used for bakery product preparation and were combined with water and ferments to obtain doughs. Doughs having a lower amount of FODMAPs were selected, and under the same conditions, cracker samples were prepared. Cooked crackers were obtained after baking in an oven at the temperature of 150 °C for 25 min.

The used ferments are described in the request of Patent n° 102022000000731 (18 January 2022): different mixes of ferments were selected, which simulated a mother yeast. The difference is that a mother yeast is characterized by a pluri-diversity of strains, while in this case, there was one type of yeast and three types of lactic bacteria. The strains were selected considering their metabolic activity versus FODMAPs carbohydrates. Doughs called “dough 1” and “dough 2” were prepared by using two mixtures of different yeasts/ferments. Each type of dough was left for different times of fermentation: 4, 6, and 8 h (Table 1).

The extraction of FODMAPs from samples was performed by following the procedure reported by Ziegler et al. [8] adapted to samples under examination: 0.4 g of samples were treated with 1 mL of methanol (MeOH) and submitted to ultrasound-assisted extraction (UAE) for 1 min. The treatment with MeOH was important to inactivate amylase and prevent starch hydrolysis during the extraction process. Then, 20 mL of milliQ water was added and extraction was performed by UAE for 5 min, followed by centrifugation at 6000 rpm for 15 min at 4 °C. The supernatant was recovered, and the residue was extracted again under the same conditions. Supernatants were combined together and filtered through 0.22 μm on Nylon filters, and then treated on OnGuard IIP cartridges (Thermo Fisher Scientific, Milan, Italy). This last purification step was important to remove possible phenolic fractions from sample matrices to improve ion chromatography performance and facilitate low-level ion analysis. The purified extracts were analyzed in duplicate after 10-fold dilution.

### 3.3. Carbohydrates Analysis

Carbohydrates analyses were performed by a DX500 series Dionex liquid chromatograph (Sunnyvale, CA, USA) equipped with an AS50 Dionex autosampler with a 25 µL loop. Chromatographic columns selected were a CarboPac PA200 (3 × 250 mm) column connected to the corresponding guard column CarboPac PA200 (3 × 50 mm), or a CarboPac PA1 (4 × 250 mm) column connected to the corresponding guard column CarboPac PA1 (4 × 50 mm). An ED50 model pulsed amperometric detector (PAD) with an Ag/AgCl reference electrode and a gold working electrode was used. Prior to usage, all eluents were degassed with helium and, during chromatographic, runs were always kept under helium to avoid carbonate ions formation. The CHROMELEON software was used for recording chromatographic data.

Qualitative analysis of FOS and/or GOS was conducted by using a CarboPac PA200 column following a previously reported method [8], slightly modified. Briefly, the mobile phase was prepared by a suitable dilution of water (Eluent A), 600 mM NaOH (Eluent B), and 500 mM sodium acetate (Eluent C). A gradient method was used (Table A1, Appendix A) for a total run time of 130 min at a flow rate of 0.25 mL/min. The injection volume was 25 µL. All available standards were analyzed in triplicate in the range of 0.25–10.0 μg/mL at seven concentration levels. All samples were analyzed in duplicate.

The qualitative and quantitative analysis of simple sugars (polyol, monosaccharides, and disaccharides) was carried out using a CarboPac PA1 following an already reported method [24], modified according to the instrument setup. Briefly, the mobile phase was prepared by a suitable dilution of water (Eluent A), 600 mM NaOH (Eluent B), and 500 mM sodium acetate (Eluent C). An isocratic method was used for the separation of polyols, mono-, and di-saccharides followed by washing and re-equilibration of the column (see Table A2, Appendix A) for a total run time of 50 min at a flow rate of 1.0 mL/min. The injection volume was 25 µL. All standards (xylitol, mannitol, sorbitol, rhamnose, galactose, glucose, fructose, sucrose) were analyzed in triplicate in the range of 0.25–10.0 μg/mL at seven concentration levels. Data were expressed as mean value (n = 2), providing standard deviation.

The limit of detection (LOD) and limit of quantification (LOQ) were calculated according to EURACHEM 2014 [25] procedure by performing ten replicates of blank samples (mobile phase) (Table A3, Appendix A). The precision was evaluated according to EURACHEM 2014 [25]: the RSD% of retention time and peak area were calculated by analyzing ten replicate measurements of the mixture of standards at a concentration of 1 μg/mL performed in one day for intra-day precision, and 30 replicates in three days for inter-day precision (Table A3, Appendix A).

In order to verify the exhaustiveness of the selected extraction method, a comparison of the results obtained from a pre-spike and a post-spike sample was performed using the internal standard L-rhamnose monohydrate at a concentration of 1 μg/mL.

## 4. Conclusions

This research was focused on a study related to food products designed for people affected by irritable bowel syndrome, and provided a useful methodology that can be applied to monitor a diet designed for the benefit of a specific pathology in order to prevent diseases.

The technological parameters which can influence the carbohydrates pattern in bakery products to be defined as low-FODMAP have been studied by analyzing different samples using HPAEC-PAD during the phases of their processing. The highly selective analytical tool for carbohydrate separation allowed us to perform the qualitative evaluation of oligosaccharides (FOS/GOS) present in samples and the quantitative determination of simple sugars. The flours with a lower number of peaks corresponding to the oligosaccharides and lower area values were selected to produce low-FODMAP products. HPAEC-PAD allowed us to study the amount of fructose compared with glucose and the number of polyols in order to choose the type of yeast/ferments and the hours of fermentation of doughs. Cooked products showed less amount of FODMAP carbohydrates than uncooked products, meaning that the degradation of sugars continues during the cooking process until the dough reaches the temperature of inactivation of ferments.

In conclusion, this work shows the high potential of the presented analytical method, which can be applied also to monitor fructooligosaccharides and inulins proposed for functional foods design; therefore, it can open new prospects in the field of research of molecules with functional features also in other fields of nutrition studies.

## Figures and Tables

**Figure 1 molecules-28-03564-f001:**
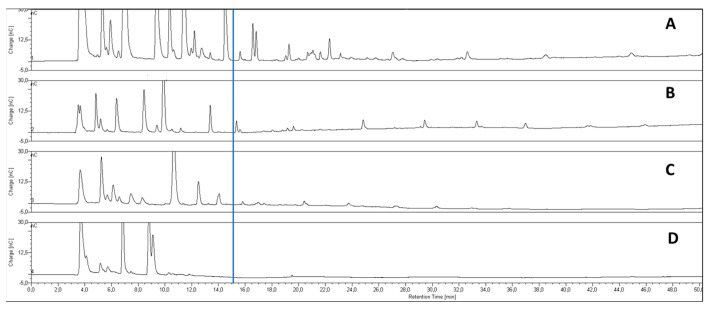
Chromatograms of flour samples. (**A**) Wheat flour; (**B**) rice flour; (**C**) emmer flour; (**D**) hemp flour.

**Figure 2 molecules-28-03564-f002:**
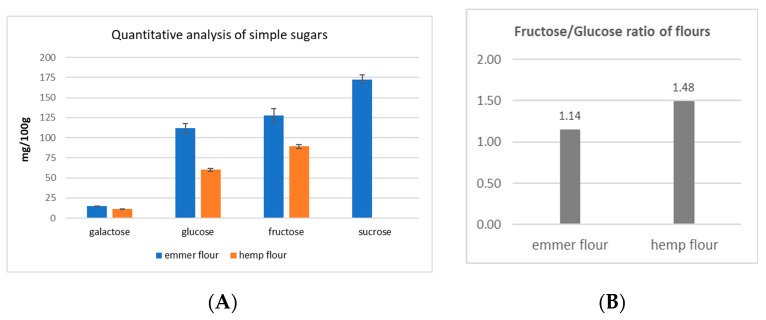
(**A**) Quantitative analysis of simple sugars in emmer and hemp flours (mg/100 g); (**B**) fructose/glucose ratio in emmer and hemp flours.

**Figure 3 molecules-28-03564-f003:**
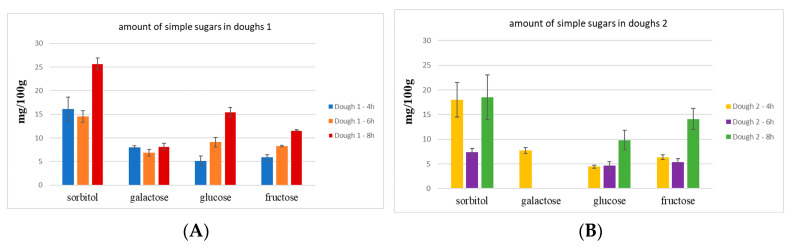
(**A**) Quantitative analysis of simple sugars in doughs 1 1; (**B**) quantitative analysis of simple sugars in doughs 2.

**Figure 4 molecules-28-03564-f004:**
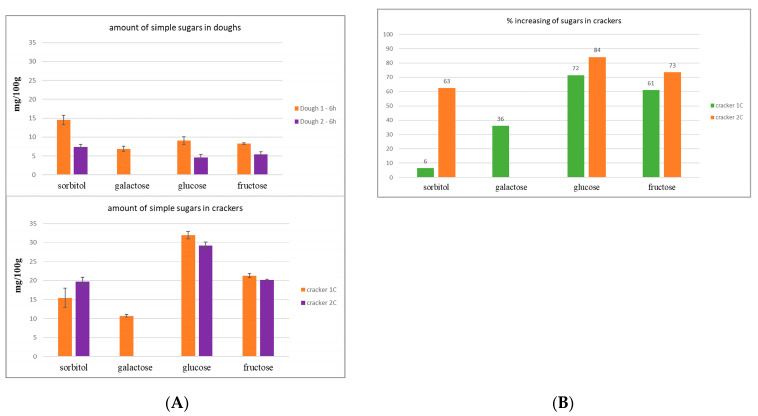
(**A**) Quantitative analysis of simple sugar in crackers in comparison with corresponding doughs; (**B**) % increase of simple sugars in crackers compared with the corresponding doughs.

**Table 1 molecules-28-03564-t001:** List of samples submitted for analysis.

Dough Samples (Ferment Type_Hours of Fermentation)	Cooked Samples
Dough 1_4 h	
Dough 1_6 h	Cracker 1C
Dough 1_8 h	
Dough 2_4 h	
Dough 2_6 h	Cracker 2C
Dough 2_8 h	

## Data Availability

Not applicable.

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
