# Peer review of "HPAEC-PAD Analytical Evaluation of Carbohydrates Pattern for the Study of Technological Parameters Effects in Low-FODMAP Food Production"

_molecules, 2023, doi:10.3390/molecules28083564_

Round 1

Reviewer 1 Report

In this manuscript of "HPAEC-PAD analytical evaluation of carbohydrates pattern for the study of technological parameters effects in low-FODMAP food production", lot of experimental work was done and it is certainly contributing to the current scientific knowledgebase. However, I am puzzled by somewhere in the paper. The following are the questions and comments of this manuscript:

1.       Why is it proposed as the only possible therapy for symptoms management in individuals who experience symptoms when eating FODMAPs?

2.       What is prebiotic function, and how does the presence of probiotics act to stimulate the growth of beneficial bacteria in the intestine?

3.       Are there other dietary interventions or therapies available for individuals with inflammatory diseases such as irritable bowel syndrome besides the low-FODMAP diet?

4.       Are there any potential risks associated with following a low-FODMAP diet, and if so, what are they?

5.       The citing references are old, please consider some new one such as https://doi.org/10.3390/molecules28041613. Or other references.

6.       What are some possible limitations or weaknesses of using HPAEC-PAD to study the technological parameters that influence the FODMAP content of bakery products?

After going through the manuscript, I feel that authors need to revise the paper to reach the level of Molecules.

Author Response

Reviewer 1 

In this manuscript of "HPAEC-PAD analytical evaluation of carbohydrates pattern for the study of technological parameters effects in low-FODMAP food production", lot of experimental work was done and it is certainly contributing to the current scientific knowledgebase. However, I am puzzled by somewhere in the paper. The following are the questions and comments of this manuscript.

R. Thank you for your comments. We believe your advises helped to improve the quality of our manuscript and we hope that the revised text is acceptable for publication.

  

  1. Why is it proposed as the only possible therapy for symptoms management in individuals who experience symptoms when eating FODMAPs? 

People affected by irritable bowel syndrome (IBS) are encouraged to follow a diet established by Monash University since there isn’t a pharmacological treatment for IBS. There are some proposed treatments, like administration of probiotics, gluten-free food consumption, prescription of medications such as antispasmodics, antidepressants, prosecretory agents, anti-diarrheal agents, antibiotics, serotonin agents, but all of them were not found to be able to cure IBS and are therefore suggested only to alleviate symptoms.  

The revised text has been enriched with such information (see introduction line 55-61) 

  1. What is prebiotic function, and how does the presence of probiotics act to stimulate the growth of beneficial bacteria in the intestine?

The prebiotic function is exerted as some carbohydrates are not hydrolyzed and absorbed in the colon, and therefore become a good substrate for the growth of bacteria of the microbiota; however, in subjects affected by IBS, the fermentation of these molecules triggers symptoms. As for probiotics, they are constituted by the bacteria themselves, so their administration helps to renew the bowel microbiota.  

The revised text has been modified to provide these details.  

  1. Are there other dietary interventions or therapies available for individuals with inflammatory diseases such as irritable bowel syndrome besides the low-FODMAP diet?

Currently the only protocol identified ad hoc is the low FODMAP diet. Australian data shows that in 75% of cases it helps to reduce symptoms. Alternatively, those suffering from IBS are provided with medical nutritional advice which includes probiotics administration, a gluten-free diet (even if they have not been diagnosed with celiac disease or suffering from gluten sensitivity), low in fiber, fat and nerve substances, such as any gastrointestinal inflammation. 

A paragraph conveying these data has been inserted in the revised text. 

  1. Are there any potential risks associated with following a low-FODMAP diet, and if so, what are they?

The low FODMAP diet is a protocol to be followed for a limited period of time in order to allow the individual to understand the tolerated quantities of one or more FODMAPs, and with the aim of scheduling a personalized food plan. A long period diet, without a balanced food plan, can lead to a lack of fibers, and therefore a potential increase in dysbiosis, of mineral salts, vitamins and an imbalance of the main nutritional principles. 

These informations have been added to the revised text.  

  1. The citing references are old, please consider some new one such as https://doi.org/10.3390/molecules28041613. Or other references.

Thank you, we updated references adding the suggested one (n.3), and also some more suggested by reviewer 2  

  1. What are some possible limitations or weaknesses of using HPAEC-PAD to study the technological parameters that influence the FODMAP content of bakery products?

The limitation of this technique is related to the limited availability of commercial standards. Indeed, there aren’t commercially available oligosaccharides with degree of polymerization higher than 5. However, on the basis of literature studies the following peaks can be identified by considering the assumption that each peak at the same distance represents a carbohydrate with one more unit. Unlikely, a quantitative evaluation of those peaks is not possible, although a semiquantitative approach can be performed by evaluation of peak areas. These considerations have been added to the text in section 2.2. 

Reviewer 2 Report

This paper describes a study which aims to explore food techniques that can be used to manipulate a group of short chain carbohydrates (that have been grouped under the acronym- FODMAP) in bakery products. FODMAPs can trigger gastrointestinal symptoms (like bloating, gas, constipation, dirrhoea and pain) associated with irritable bowel syndrome (IBS).  FODMAPs (Fermentable Oligo- Di- and Mono-saccharides And Polyols) include: lactose, excess fructose (that is in excess of glucose), fructans/fructo-oligosaccharides (FOS), galacto-oligosaccharides (GOS- stachyose, raffinose), sorbitol and mannitol.   Fructans are one of the major FODMAPs present in our diet and the major type of FODMAP present in grains/cereals products.

In this paper, two major approaches to reduce the FODMAP content were explored including choice of ingredient (ie. type of flour) and length of fermentation time.    Techniques use to assess the efficacy of these different processing approaches were HPAEC-PAD for the qualitative assessment of changes in FOS (eg. kestose and nystose) and HPLC for quantification of sugars – glucose, fructose, sucrose, sorbitol, mannitol and galactose.

The results do confirm that choice of ingredient (flour) and use of length of ferment can indeed be used to manipulate FODMAPs in bakery products.

Comments:

Abstract

1      Abstract – The  low FODMAP diet therapy was developed for managing IBS symtpoms.  For patients with inflammatory bowel disease (IBD) – it is only if these individuals also experience concurrent IBS symptoms that the low FODMAP diet may be used. Also, IBS is not considered a disease – rather a disorder.

Introduction -

1.     Maltodextrins – while these are oligosaccharides, maltodextrins are not considered ‘fibres’ as generally the body easily digests these  (unless modified in some way to produce a resistant maltodextrins).  Maltodextrins are often produced following the hydrolysis of starch.

2.     From ‘vegetable origin’- it may be more accurate to say plant origin.

Results / discussion

3.     Results are given as mean,  SEM or SD?  How many replicates?

4.    Section 2.2 – selection of flours-  clarify what you mean by ‘grain’ flour? Are you referring to wheat?  Replace ‘grain’ with ‘wheat throughout the paper.

5.    HPAEC-PAD chromatographs.  A lengthy explanation has been given for the use of methanol to extract the carbohydrates.  However, there is no identification of the various peaks for the extracted flours shown in the chromatograms (see Fig 1).   Identification and quantification of the peaks is essential to accurately assess the efficacy of these various technical approaches.    This is a major limitation of this study. You have presented qualitative and not assessment.  It is hard to conclude that the use of emmer flour can assist in lowering levels of oligosaccharides as no peaks have been identified or quantified.

6.    Quantification was instead carried out using HPLC using different columns and mobile phase for the sugars glucose, fructose, galactose ad sucrose.  Fig 2. It appears that they are using fructose/ glucose ratio and galactose as a surrogate makers for the break down of fructan and GOS in the fermented/ baked product.

Section 2.3

7.    …..By choosing the right mix of ferments and the right time of fermentation it is possible to find the 180 compromise to ensure that each class of carbohydrates respect the cut-off limit established by Monash University [11].    However these cutoff levels established by the Monash team are all based on a ‘typical as eaten serve’ and not raw dough. So your discussion around cut-offs needs to be confided to the end cracker product (as eaten) and a typical serve of that cracker.

8.    The study that explored using the different yeast/ferments plus the different length of fermentation is confusing. Information shown in Table 1 needs to be clearer  – is it to be assumed that dough 1 and dough 2 are the different ferments that were the left for different length of times 4h and 6h?  Why not compare the 4 flours at the same 3 time points 4, 6, 8h?

9.    The results shown in Fig 3- the chromatograms – were very difficult to interpret.  There has been no identification or quantification of the peaks.

10. A major problem with this study is that you need to quantitate the levels of fructans in the product – the approach used here with the HPAEC-PAD  does not allow for the quantitification of the major type of FODMAP present in grains/flours- fructan.  The papers quoted in the text (ref 13) uses an enzyme kit to quantify total fructan – so why was this method not used this current study?  Certainly the HPAEC-PAD –  does provide some interesting qualitative data, however, quantify fructans is vital if you are to work comply with the ‘cut-off’ limits that are used in this area.

11. Fig 5 – it is interesting that you are measuring polyols (sorbitol) as usually it is mannitol that increases in these types of fermentation.  Are you sure that it was sorbitol and not mannitol?

12.  Some published research/aritcles in this area have not been referred to;

Laatikainen, R., Koskenpato, J., Hongisto, S.M., Loponen, J., Poussa, T., Hillilä, M., Korpela, R., 2016. Randomised clinical trial: low-FODMAP rye bread vs. regular rye bread to relieve the symptoms of irritable bowel syndrome. Aliment. Pharmacol. Ther. 44, 460–470.

Struyf, N., Laurent, J., Verspreet, J., Verstrepen, K.J., Courtin, C.M., 2017. Saccharomyces cerevisiae and Kluyveromyces marxianus cocultures allow reduction of fermentable oligo-, di-, and monosaccharides and polyols levels in whole wheat bread. J. Agric. Food Chem. 65, 8704–8713.

Nilsson, U., Öste, R., Jägerstad, M., 1987. Cereal fructans: hydrolysis by yeast invertase, in vitro and during fermentation. J. Cereal Sci. 6, 53–60.

Muir JG, Varney JE, Ajamian M, Gibson PR.  2019. Gluten-free and low-FODMAP sourdoughs for patients with coeliac disease and irritable bowel syndrome: A clinical perspective. Int J Food Microbiol. 290, 237-246.

Author Response

Thank you for your comments. We believe your advices helped to improve the quality of our manuscript and we hope that the revised text is acceptable for publication.

See attached text for punctual responses.

Reviewer 3 Report

1. What are technological parameters effects?

2. Abstract: The authors only describe the background and what did they do. However, no information on the results and conclusion was presented. The structure of abstract should be "background-method-results and conclusion"

3. Introduction: paragraph 1: and pectins, deleted and; thus, increasing...: deleleted ,  ; paragraph 5: Check the first sentence; 

4.  Figures: All chromatograms should be redrawn. Improve figure resolution.

5. The whole manuscript is obscure and difficult to understand

Author Response

  1. What are technological parameters effects?

R. As reported in literature, technological parameters that can influence the carbohydrate pattern in low-FODMAP products are: a) choice of flours; b) choice of ferment/yeast; c) time of fermentation; d) pH of doughs ; e) temperature of fermentation of doughs; f) cooking processing. We inserted some lines in the revised text (Introduction) in order to explain their possible effects, and how our approach can be proposed to select the opportune conditions to achieve low-FODMAP products.

  1. Abstract: The authors only describe the background and what did they do. However, no information on the results and conclusion was presented. The structure of abstract should be "background-method-results and conclusion"

R. Thank you for your suggestion. We divided the abstract in background-method-results and conclusion in order to improve its quality.

  1. Introduction: paragraph 1: and pectins, deleted and; thus, increasing...: deleleted ,  ; paragraph 5: Check the first sentence;

R. We corrected the manuscript text according to your advice.

  1. Figures: All chromatograms should be redrawn. Improve figure resolution.

R. Figures have been replaced with images at higher resolution.

  1. The whole manuscript is obscure and difficult to understand

R. We have rewritten some parts of the text with the aim of improving clarity, and hope that the revised version is easier to understand.

Round 2

Reviewer 2 Report

The major limitation with this study is that you need to quantify the levels of the major type of FODMAP present in baked products -  fructans.  The approach used here involving  use of HPAEC-PAD - does not provide this information.   There are enzymatic kits that are available for the analysis of total fructans and these should have been used. 

Author Response

The major limitation with this study is that you need to quantify the levels of the major type of FODMAP present in baked products -  fructans.  The approach used here involving  use of HPAEC-PAD - does not provide this information.   There are enzymatic kits that are available for the analysis of total fructans and these should have been used.

R. Thank you for your suggestion. Fructans in our case are not the major type of FODMAP since in the final products they were not detected thank to the proposed approach that allowed to follow the effects of the technological parameters. HPAEC PAD cannot provide a punctual quantitative evaluation, but can detect if those molecules are present, and allows to follow the possible decrease of each oligosaccharide by considering the values of peak area. Moreover, when necessary, a quantitative analysis on oligosaccharides can be performed by using calibration curve of nystose, as proposed in some previous works.

Enzimatic kits can of course be of help and used for samples containing fructans. However, in this study, oligosaccharides were not detected in final products, and therefore the use of an enzymatic kit cannot add any other useful information. Indeed, only the quantitative analysis of simple sugars was performed (paragraph 2.4). The text has been modified to convey this information.